

# Hsa_circ_0009096/miR-370-3p modulates hepatic stellate cell proliferation and fibrosis during biliary atresia pathogenesis

Zhouguang Wu, Bin Wang, Siqi Chen, Taoyan Zuo, Wenjie Zhang, Zhen Cheng, Jingru Fu and Jiafeng Gong

Department of General Surgery, Shenzhen Children's Hospital, Shenzhen, China

Corresponding author
Bin Wang, szwb1967@126.com

## ABSTRACT

**Background:** Hepatic stellate cell (HSC) activation and hepatic fibrosis mediated biliary atresia (BA) development, but the underlying molecular mechanisms are poorly understood. This study aimed to investigate the roles of circRNA hsa_circ_0009096 in the regulation of HSC proliferation and hepatic fibrosis.

**Methods:** A cellular hepatic fibrosis model was established by treating LX-2 cells with transforming growth factor β (TGF-β1). RNaseR and actinomycin D assays were performed to detect hsa_circ_0009096 stability. Expression of hsa_circ_0009096, miR-370-3p, and target genes was detected using reverse transcription-qPCR. Direct binding of hsa_circ_0009096 to miR-370-3p was validated using dual luciferase reporter assay. Cell cycle progression and apoptosis of LX-2 cells were assessed using flow cytometry. The alpha-smooth muscle actin (α-SMA), collagen 1A1 (COL1A1), and TGF beta receptor 2 (TGFBR2) protein levels in LX-2 cells were analyzed using immunocytochemistry and western blotting.

**Results:** Hsa_circ_0009096 exhibited more resistance to RNase R and actinomycinD digestion than UTRN mRNA. Hsa_circ_0009096 expression increased significantly in LX-2 cells treated with TGF-β1, accompanied by elevated α-SMA and COL1A1 expression. Hsa_circ_0009096 siRNAs effectively promoted miR-370-3p and suppressed TGFBR2 expression in LX-2 cells, mediated by direct association of hsa_circ_0009096 with miR-370-3p. Hsa_circ_0009096 siRNA interfered with the cell cycle progression, promoted apoptosis, and reduced α-SMA and COL1A1 expression in LX-2 cells treated with TGF-β1. MiR-370-3p inhibitors mitigated the alterations in cell cycle progression, apoptosis, and α-SMA, COL1A1, and TGFBR2 expression in LX-2 cells caused by hsa_circ_0009096 siRNA. In conclusion, hsa_circ_0009096 promoted HSC proliferation and hepatic fibrosis during BA pathogenesis by accelerating TGFBR2 expression by sponging miR-370-3p.

## INTRODUCTION

Biliary atresia (BA) is a severe neonatal liver disorder characterized by progressive obstruction of the extrahepatic biliary tree of infants, which is usually accompanied by rapid development of liver fibrosis and cholestasis during the first months after birth (*Lakshminarayanan & Davenport, 2016*; *Wehrman, Waisbourd-Zinman & Wells, 2019*).

BA incidences vary drastically according to region, and a high incidence of over 1 per 5,000 was recorded in certain Asian regions, which could commonly lead to neonatal jaundice (*Vij & Rela, 2020*). BA etiology is reportedly associated with multiple factors, including genetic variation, virus infections, microchimerism, exposure to environmental toxins, immunologic injury, and primary vascular anomalies (*Vij & Rela, 2020*). Infants with BA commonly present with hyperbilirubinemia, dark urine, hepatosplenomegaly, acholic stool, and progressive liver failure, potentially developing into fibrosis, portal hypertension, end-stage liver disease, and sometimes, death within two years without proper treatment (*Bezerra et al., 2018*; *Wehrman, Waisbourd-Zinman & Wells, 2019*; *Vij & Rela, 2020*). The recent clinical management of patients with BA greatly depends on the application of Kasai hepatic portoenterostomy (KPE) to remove the extrahepatic biliary system, restore bile flow, and relieve the obstruction, thereby promoting the survival of infants with BA combined with liver transplantation (*Bezerra et al., 2018*). However, early diagnosis, progressive liver fibrosis prevention, and long-term complication management in patients with BA remain great medical challenges, partially due to the poorly understood pathogenetic mechanisms (*Bezerra et al., 2018*).

Hepatic stellate cells (HSCs), which were first characterized in 1876, are essential resident liver cells mainly distributed in the subendothelial space of Disse and serve as a major reservoir of retinyl esters in the form of cytoplasmic lipid droplets (*Tsuchida & Friedman, 2017*). In normal physiological environments, HSCs are commonly in the quiescent and non-proliferative state in hepatic tissues (*Tsuchida & Friedman, 2017*; *Dewidar et al., 2019*). Under pathogenic conditions such as acute or chronic liver injuries, which usually induce the release of TGF-β (transforming growth factor β), HSCs can be activated and transdifferentiated from the quiescent fat-storing cells into proliferative myofibroblasts with enhanced expression of the intracellular microfilament protein alpha-smooth muscle actin (α-SMA), collagens, and other extracellular matrix (ECM) proteins (*Dewidar et al., 2019*). HSC activation and transformation have been characterized as critical mediating factors of hepatic fibrosis, which has been linked with the pathogenesis of various hepatic conditions, including alcoholic liver disease, cirrhosis, nonalcoholic fatty liver disease (NAFLD), and liver cancer (*Tsuchida & Friedman, 2017*; *Dewidar et al., 2019*; *Yang et al., 2021*). More importantly, previous studies have demonstrated that HSC activation is critically associated with hepatic fibrogenesis during BA pathogenesis (*Xiao et al., 2014*; *Vij & Rela, 2020*). Furthermore, HSC activation and α-SMA expression could be used as reliable biomarkers for predicting the outcome of patients with BA undergoing KPE (*Shteyer et al., 2006*; *Dong, Luo & Zheng, 2012*). However, the molecular mechanism modulating HSC activation associated with BA development and progression remains poorly elucidated.

Circular RNAs (circRNAs), a large set of non-coding RNAs produced through the back-splicing of gene pre-mRNAs, play important biological roles and have been established as key regulators of human disorders (*Haddad & Lorenzen, 2019*; *Kristensen et al., 2019*; *Chen, 2020*). Extensive research has shown that circRNAs perform their potent regulatory functions by mainly acting as competing endogenous RNAs (ceRNAs) or protein sponges, as well as scaffolds for stabilizing circRNA-protein complexes, thus

fundamentally altering various cellular processes, including cell proliferation and transformation (*Chen, 2020*). Moreover, the differential expression of circRNAs has been shown to contribute to HSC activation and ECM deposition associated with liver fibrosis progression (*Zhou et al., 2018*; *Riaz & Li, 2019*; *Zhou et al., 2019*). Our project team identified a large number of differentially expressed circRNAs for the first time in liver tissues of patients with BA and confirmed that has_circ_0009096 was significantly upregulated in liver tissues of patients with BA (*Zhang et al., 2023*). We also found that has_circ_0009096 could regulate autophagy through the miR-483-3p/IGF-1 axis to promote HSC fibrosis in BA development (*Liu et al., 2023*). Because of its complex regulatory network, more regulatory mechanisms have yet to be deeply explored. In this study, we explored the roles of hsa_circ_0009096 in the regulation of HSC proliferation, apoptosis, and ECM deposition, as well as its interaction with downstream microRNAs to regulate target gene expression. These investigations aim to provide new insights into the hsa_circ_0009096-mediated mechanisms driving HSC activation and BA pathogenesis.

## MATERIALS AND METHODS

### Cell lines and culture

LX-2 (#CC4023) and HEK-293T (#CC4003) cells were obtained from CellCook (Guangzhou, China). LX-2 and HEK-293T cells were cultured in Dulbecco's modified Eagle's medium (DMEM; #C11995500BT; Thermo Fisher Scientific, Waltham, MA, USA) supplemented with 10% fetal bovine serum (#26010074; Thermo Fisher Scientific, Waltham, MA, USA) and 1% penicillin/streptomycin (#15140122; Thermo Fisher Scientific, Waltham, MA, USA) at 37 °C in a humidified cell culture chamber with a supply of 5% $CO_2$. To induce fibrosis of LX-2 cells, these cells were cultured in DMEM containing 0, 0.5, or 2-ng/mL rhTGF-β1 (Recombinant Human TGF β1; #BPR147; Bersee, Beijing, China) under normal conditions, as previously described (*Luo et al., 2019*).

### RNA interference

For suppression of hsa_circ_0009096 expression in LX-2 cells, siRNA-1 (5′-GAGAATGGTTTGATGCTATAA-3′), siRNA-2 (5′-AACATCTAGAGAATGG TTTGA-3′), siRNA-3 (5′-CTAGAGAATGGTTTGATGCTA-3′), and negative control (NC) (5′-TTCTCCGAACGTGTCACGTTT-3′) were prepared by Genepharma (Shanghai, China). Similarly, to repress the miR-370-3p expression in cultured LX-2 cells, the miR-370-3p (5′-ACCAGGUUCCACCCCAGCAGGC-3′) and negative control sequence inhibitors (NC: 5′-CAGUACUUUUGUGUAGUACA A-3′) were also prepared by Genepharma. The above mentioned siRNA or microRNA inhibitors were introduced into the LX-2 cells using the Invitrogen Lipofectamine 3000 Transfection Reagent (#L3000150; Thermo Fisher Scientific, Waltham, MA, USA) following the manufacturer's instructions. The expression-suppressing efficacy was then evaluated using reverse transcription-qPCR (RT-qPCR) 24 h after transfection.

## RNase R treatment and the actinomycin D assay

A total of 2 μg of total RNA was incubated for 8 min at 37 °C with or without 6 U RNase R (#R0300; Geneseed, Guangzhou, China), subsequently deactivated with RNase R at 70 °C for 10 min, and then analyzed using RT-qPCR. Actinomycin D (#50-76-0, APExBIO, Houston, TX, USA) at a concentration of 1 μM was applied to LX-2 cells at specified time intervals (0, 12, and 24 h). Following cell harvesting, total RNA was extracted for RT-qPCR. The PCR products were detected using agarose gel electrophoresis.

## Reverse transcription-qPCR

The expression levels of circRNAs, microRNAs, and mRNAs of interest in the cultured LX-2 cells were determined using RT-qPCR. Briefly, total RNA was extracted from the cultured LX-2 cells using the Trizol kit (#B511311; Sangon Biotech, Shanghai, China) following the manufacturer's instructions. To avoid contamination by genomic DNA, the samples were treated with a DNase kit (Ambion, Austin, TX, USA). Reverse transcription without reverse transcriptase was also performed to assess gDNA contamination.
The RNA quantity and purity (A260/A280) were evaluated using Nano-100 (ALLSHENG, Hangzhou, China). Then, the cDNA samples were synthesized from approximately 1.5-μg RNA samples of each group using the EasyScript RT kit (#AE101-02; Transgen Biotech company, Beijing, China) following the manufacturer's instructions and stored at −20 °C for further experiments. Subsequently, quantitative PCR assay was performed on the ABI Stepone Plus instrument (Applied Biosystems, Foster City, CA, United States) to measure the relative expression of circRNA, microRNA, or mRNAs using the TransStart qPCR kit (#AQ101-01; Transgen Biotech company, Beijing, China) following the manufacturer's instructions. The expression levels were quantitated *via* the standard $2^{(-\Delta\Delta Ct)}$ calculation method using U6 or GAPDH as reference genes. The qPCR for each group comprised three replicates. The information on the primers used in expressional quantitation is listed in Table 1. All primers were synthesized by GenePharma and were PAGE-purified.

## Dual luciferase reporter assay

The association of hsa_circ_0009096 with hsa-miR-370-3p in 293T cells was validated through the dual luciferase reporter assay using the Nano-Glo Dual-Luciferase® Reporter kit (#N1630; Promega, Madison, WI, USA) following the manufacturer's instructions. Briefly, the circ_0009096 wild-type or mutant sequences were, respectively, ligated with the pmirGLO vectors, which were induced into 293T cells using the Lipofectamine 3000 kit as introduced above, along with the hsa-miR-370-3p mimics (5′-GCCUGCUGGGGUGGAACCUGGU-3′) or its negative control sequences (5′-UUCUCCGAACGUGUCACGUTT-3′). Finally, the cultured 293T cells were lysed 48 h after transfection, and cell lysates were used to measure luciferase activities using a GloMax luminometer. The binding of circ_0009096 with hsa-miR-370-3p was assessed by alterations of luciferase activities between different groups.

**Table 1 The sequences of primer pairs used for RT-q PCR.**

| Primer ID | Primer sequences (5′–3′) |
| --- | --- |
| hsa_circ_0009096-F | TTCGATAGCTTTCTGGGCCG |
| hsa_circ_0009096-R | CTCTGTGGTCTGAATGGCAGT |
| H-α-SMA-F | GCAGGGTGGGATGCTCTT |
| H-α-SMA-R | GGTGATGGTGGGAATGGG |
| H-COL1A1-F | GCCAAGACGAAGACATCCCA |
| H-COL1A1-R | GGCAGTTCTTGGTCTCGTCA |
| H-TGFBR2-F | GTAGCTCTGATGAGTGCAATGAC |
| H-TGFBR2-R | CAGATATGGCAACTCCCAGTG |
| H-SMAD4-F | CTCATGTGATCTATGCCCGTC |
| H-SMAD4-R | AGGTGATACAACTCGTTCGTAGT |
| H-TGFB2-F | CAGTGGGAAGACCCCACATC |
| H-TGFB2-R | CAATAGGCCGCATCCAAAGC |
| H-THBS1-F | GCCATCCGCACTAACTACATT |
| H-THBS1-R | TCCGTTGTGATAGCATAGGGG |
| H-GAPDH-F | GAGTCAACGGATTTGGTCGT |
| H-GAPDH-R | GACAAGCTTCCCGTTCTCAG |
| H-UTRN-F | GCAAATGGGAAAAGAAGGCCT |
| H-UTRN-R | GGACTGCCGGGAAGATTCAG |
| hsa-miR-21-5p-RT | GTCGTATCCAGTGCAGGGTCCGAGGTATTCGCACTGGATACGACTCAACA |
| hsa-miR-21-5p-F | TAGCTTATCAGACTGATG |
| hsa-miR-182-5p-F | TTTggcaaTggTagaacTca |
| hsa-miR-182-5p-RT | GTCGTATCCAGTGCAGGGTCCGAGGTATTCGCACTGGATACGACagtgtg |
| hsa-miR-370-3p-RT | GTCGTATCCAGTGCAGGGTCCGAGGTATTCGCACTGGATACGACACCAGG |
| hsa-miR-370-3p-F | GCCTGCTGGGGTGGAACCT |
| hsa-miR-421-RT | CTCAACTGGTGTCGTGGAGTCGGCAATTCAGTTGAGGCGCCCAA |
| hsa-miR-421-F | GCCGAGATCAACAGACATTAA |
| hsa-U6-F | CTCGCTTCGGCAGCACA |
| hsa-U6-R | AACGCTTCACGAATTTGCGT |

## Flow cytometry for cell cycle progression analysis

Cell cycle progression in the cultured LX-2 cells was determined using flow cytometry with the Cell Cycle Analysis Kit (#ab287852; Abcam, Cambridge, UK) following the manufacturer's protocol. Briefly, LX-2 cells cultured in a six-well plate ($5 \times 10^5$ cells/well) were harvested *via* centrifugation at 400 g for 4 min, washed with pre-chilled cell cycle assay buffer, fixed by incubation with 70% ethanol on ice for 40 min, washed again with 2-ml cell cycle assay buffer, incubated with 450-ul staining solution in the dark for 40 min, and finally analyzed using a flow cytometer. The LX-2 cell cycle progression was detected through at least three biological replicates of flow cytometry assays.

## Flow cytometry for cell apoptosis analysis

Apoptosis of the LX-2 cells was analyzed using flow cytometry following treatment with the Annexin V-PI Apoptosis Staining kit (#ab214484; Abcam, Cambridge, UK) according to the manufacturer's instructions. Briefly, the cultured LX-2 cells ($1 \times 10^6$ cells) after designated transfections and treatments were harvested through centrifugation at 400 g for 5 min, resuspended in 1X Annexin-binding buffer (100 ul), and incubated with Annexin V (5 μL) and PI (5 μL) solutions in the dark at room temperature for 14 min. Following incubation, the cell mixtures were then mixed with 400 μL 1X Annexin-binding buffer and subjected to flow cytometry. Three biological replicates were performed to assess the percentages of apoptotic LX-2 cells.

## Immunocytochemistry (ICC) staining

*In situ* collagen 1A1 (COL1A1) and α-SMA protein levels in the cultured LX-2 cells following specified transfections and treatments were detected using the ICC method. Briefly, LX-2 cell slides were first fixed *via* incubation with paraformaldehyde (4%) for 12 min at room temperature, washed thrice with PBS solution, permeabilized with PBS solution containing 0.1 Triton X-100 for 8 min, blocked with 1% BSA solution for 30 min at room temperature, incubated with diluted primary antibodies targeting COL1A1 (#8784; Santa Cruz, California, USA; 1:50) or α-SMA (#14395-1-Ap; ProteinTech, IL, US; 1:500) overnight at 4 °C, washed thrice times with PBS solution for 6 min, and incubated with diluted secondary antibodies for 1 h. After counter-staining with 0.2 μg/mL DAPI for 1 min, the LX-2 cell slides were then developed with HRP substrates, and then mounted on and observed under a microscope. ICC scores were calculated using the following formula: ICC score = intensity score × percentage score. The intensity score was measured according to the intensity of staining (0: negative, 1: light-yellow particle, 2: brown-yellow particle, and 3: brown particle). The percentage score was determined using the percentage of stained area (0: 0%, 1: <10%, 2: 10–49%, 3: 50–74%, and 4: 75–100%) (*Guo et al., 2021*).

## Western blotting (WB)

The LX-2 cells were lysed using $1 \times$ RIPA buffer (#89901; ThermoFisher Scientific, Waltham, MA, USA) for 20 min on ice. Following the manufacturer's instructions, the supernatant was collected after centrifugation at 12,000 g for 10 min. The Pierce™ BCA Protein Assay Kit (#23225; Thermo Scientific, Waltham, MA, USA) was then used to quantify the protein. Thirty micrograms of protein were separated using 8–10% SDS-PAGE and then transferred to a PVDF membrane. After the membrane was cleaned with $1 \times$ TBST, it was allowed to soak for 1 h at room temperature in 5% non-fat milk. Subsequently, the membrane was incubated at 4 °C overnight with the corresponding primary antibodies targeting COL1A1 (#ab34710; Abcam; 1:1,000; Cambridge, UK), α-SMA (#23081-1-AP; Proteintech, IL, USA; 1:1,000), TGFBR2 (#BA0526-2; Boster, California, US; 1:1,000), or GAPDH (#60004-1-lg; Proteintech, Rosemont, IL, USA; 1:100,000). The membrane was then treated for 1 h at room temperature with a secondary antibody. The target protein bands were visualized using a chemiluminescent western blot detection kit (#32209, Thermo Fisher Scientific, Waltham, MA, USA). ImageJ software

(NIH, Bethesda, MD, USA) was used to analyze the greyscale value of each band. The relative intensity was calculated by dividing the normalized target protein/GAPDH ratio of the test samples by that of the control samples.

## Statistical analysis

Each experiment in this study was repeated three times independently. All quantitative data are presented as mean ± standard deviation. Student's t-test or one- and two-way analysis of variance (ANOVA) followed by Dunnett's multiple-comparison posttest were performed to assess the statistical significance using Graphpad Prism (v6.0) software. NS, not significant ($p > 0.05$), *$p < 0.05$, **$p < 0.01$.

## RESULTS

### Hsa_circ_0009096 was resistant to RNase R and actinomycin D treatment

Hsa_circ_0009096 was characterized using PCR with divergent primers followed by agarose gel electrophoresis and Sanger sequencing in our previous study (*Zhang et al., 2023*), which confirmed the specialty and back-spliced junction of hsa_circ_0009096. In this study, we performed additional RNase R and actinomycin D assays on the LX-2 cells. After RNase R and actinomycin D treatment, the linear RNAs reduced significantly, but the circular RNAs were more stable and resisted RNase R (Fig. 1A) and actinomycin D (Fig. 1B), further verifying the circular nature of hsa_circ_0009096.

### Elevated hsa_circ_0009096 expression in the induced hepatic fibrosis model

To investigate the functions of circRNAs in BA development, we previously conducted RNA sequencing to identify differentially expressed circRNAs in liver tissues collected from patients with BA, using those from patients with congenital choledocho cysts as the control group. The expression of hsa_circ_0009096 was significantly elevated in the hepatic tissues of patients with BA (*Zhang et al., 2023*). To investigate the effect of hsa_circ_0009096 on hepatic fibrosis in BA, we established a hepatic fibrosis model by treating LX-2 cells with different TGF-β1 concentrations. Using an RT-qPCR assay, we demonstrated that α-SMA and COL1A1 mRNA expression in the LX-2 cells concentration-dependently increased after TGF-β1 treatment (Figs. 2A and 2B). Furthermore, the WB results were consistent with the RT-qPCR data (Fig. 2C). These results suggested that fibrosis was effectively induced in the cultured LX-2 cells. Interestingly, we also confirmed *via* the RT-qPCR method that TGF-β1 treatment significantly upregulated the hsa_circ_0009096 expression in LX-2 cells (Fig. 2D), suggesting the potential functions of hsa_circ_0009096 in liver fibrosis during BA development.

### Hsa_circ_0009096 sponges miR-370-3p to modulate TGFBR2 expression in LX-2 cells

To explore of the potential roles of hsa_circ_0009096 in liver fibrosis in BA development and its downstream targets, we silenced hsa_circ_0009096 expression in the LX-2 cells by

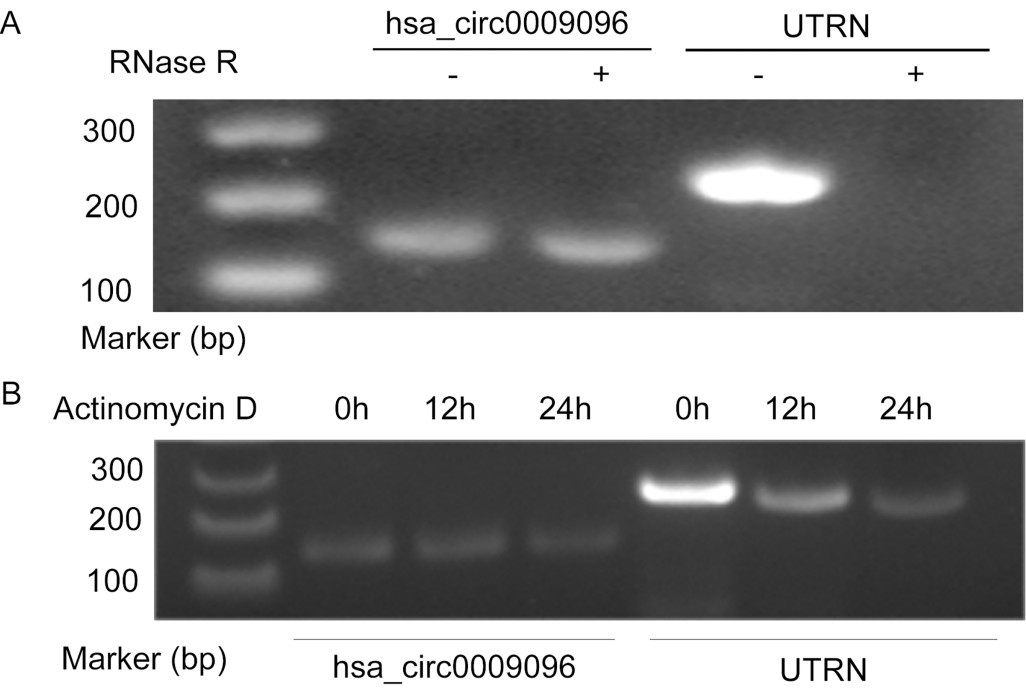

**Figure 1 Hsa_circ_0009096 was resistant to RNase R and actinomycin D treatment.** (A) Agarose gel showing hsa_circ_0009096 was resistant to RNase R digestion compared with linear UTRN in LX-2 cells ($n$ = 3 biological replicate per group). (B) Agarose gel showing hsa_circ_0009096 was resistant to actinomycin D treatment compared with linear UTRN in LX-2 cells ($n$ = 3 biological replicate per group).

transfecting them with three different siRNAs targeting hsa_circ_0009096. The following RT-qPCR assay validated the significant decrease in hsa_circ_0009096 expression in the LX-2 cells by the three siRNAs compared with the NC group (Fig. 3A). The LX-2 cells that were transfected with siRNA-2 showed the greatest decrease in hsa_circ_0009096 expression, which was used for the following assay.

According to our previous prediction of the hsa_circ_0009096-miRNA-TGFβ signaling pathway based on the ceRNA mechanism (*Zhang et al., 2023*), we selected four target genes TGF beta receptor 2 (*TGFBR2*), Mothers Against Decapentaplegic Homolog 4 (*SMAD4*), Transforming Growth Factor Beta 2 (*TGFB2*), and Thrombospondin 1 (*THBS1*) for further study based on the following criteria: (1) the target genes are located in the middle and upper reaches of the TGFβ signaling pathway and activate the pathway upon upregulation; (2) multiple miRNAs are likely to regulate the target genes. Using RT-qPCR, we confirmed that the *TGFBR2* expression was remarkably suppressed in LX-2 cells by the transfection with hsa_circ_0009096 siRNA in contrast to the negative control group (Fig. 3B). Next, we screened miR-370-3p and miR-21-5p, which target *TGFBR2*, for validation (*Li et al., 2022*; *Wang et al., 2023*). The results showed that the expression of miR-370-3p, but not the miR-21-5p, was then greatly elevated in LX-2 cells transfected with hsa_circ_0009096 siRNA (Fig. 3C). Finally, we performed a dual luciferase reporter assay, which showed that the luciferase activity in the cultured 293T cells expressing the wild-type hsa_circ_0009096 sequence #1 was significantly repressed by miR-370-3p

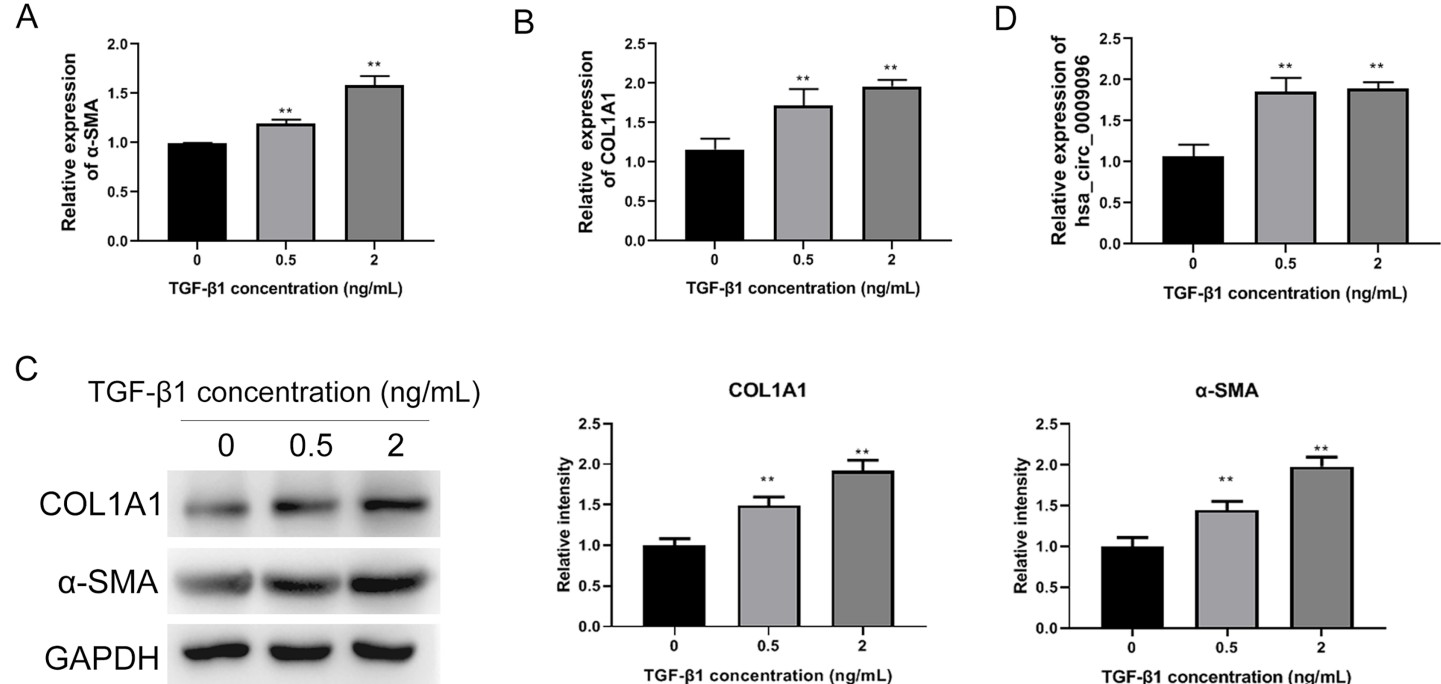

**Figure 2 Elevated hsa_circ_0009096 expression in the induced hepatic fibrosis model.** LX-2 cells were treated with 0, 0.5, or 2 ng/ml TGF-β for 24 h. The mRNA expression of α-SMA (A) and COL1A1 (B) was detected by RT-qPCR ($n$ = 3 biological replicates per group). (C) The protein expression of α-SMA and COL1A1 was accessed by western blotting ($n$ = 3 biological replicates per group). (D) The circRNA expression of hsa_circ_0009096 was evaluated by RT-qPCR ($n$ = 3 biological replicates per group). α-SMA: alpha-smooth muscle actin; COL1A1: collagen 1A1; TGF-β1: transforming growth factor beta 1; **$P < 0.01$.

mimics, but not in cells expressing the mutant hsa_circ_0009096 sequence (Fig. 3D). No alteration of luciferase activity was observed in the 293T cells expressing the wild-type hsa_circ_0009096 sequence #2 (Fig. 3D). These results proved that hsa_circ_0009096 binds to miR-370-3p at predicted site #1 to regulate *TGFBR2* expression in LX-2 cells.

## Hsa_circ_0009096 silencing suppressed TGF-β1-induced HSC proliferation and fibrosis

We analyzed the influence of hsa_circ_0009096 silencing on HSC proliferation and fibrosis using the cellular model established with TGF-β1 treatment. We observed that hsa_circ_0009096 siRNA significantly reduced the COL1A1 and a-SMA mRNA expression in the TGF-β1-treated LX-2 cells compared with the siNC + TGF-β1 group (Figs. 4A and 4B). Moreover, the *TGFBR2* mRNA expression levels in the LX-2 cells were greatly promoted following the TGF-β1 treatment, which was also effectively decreased by hsa_circ_0009096 siRNA (Fig. 4C). In contrast, the miR-370-3p expression in LX-2 cells was drastically suppressed by TGF-β1 treatment, which was then significantly recovered by hsa_circ_0009096 siRNA (Fig. 4D). Furthermore, flow cytometry showed that TGF-β1 promoted cell cycle progression from the S phase to the G2/M phases and that hsa_circ_0009096 siRNA significantly induced cell cycle arrest at the G1 stage in the LX-2 cells (Fig. 4E). Hsa_circ_0009096 siRNA partially counteracted the inhibition of apoptosis

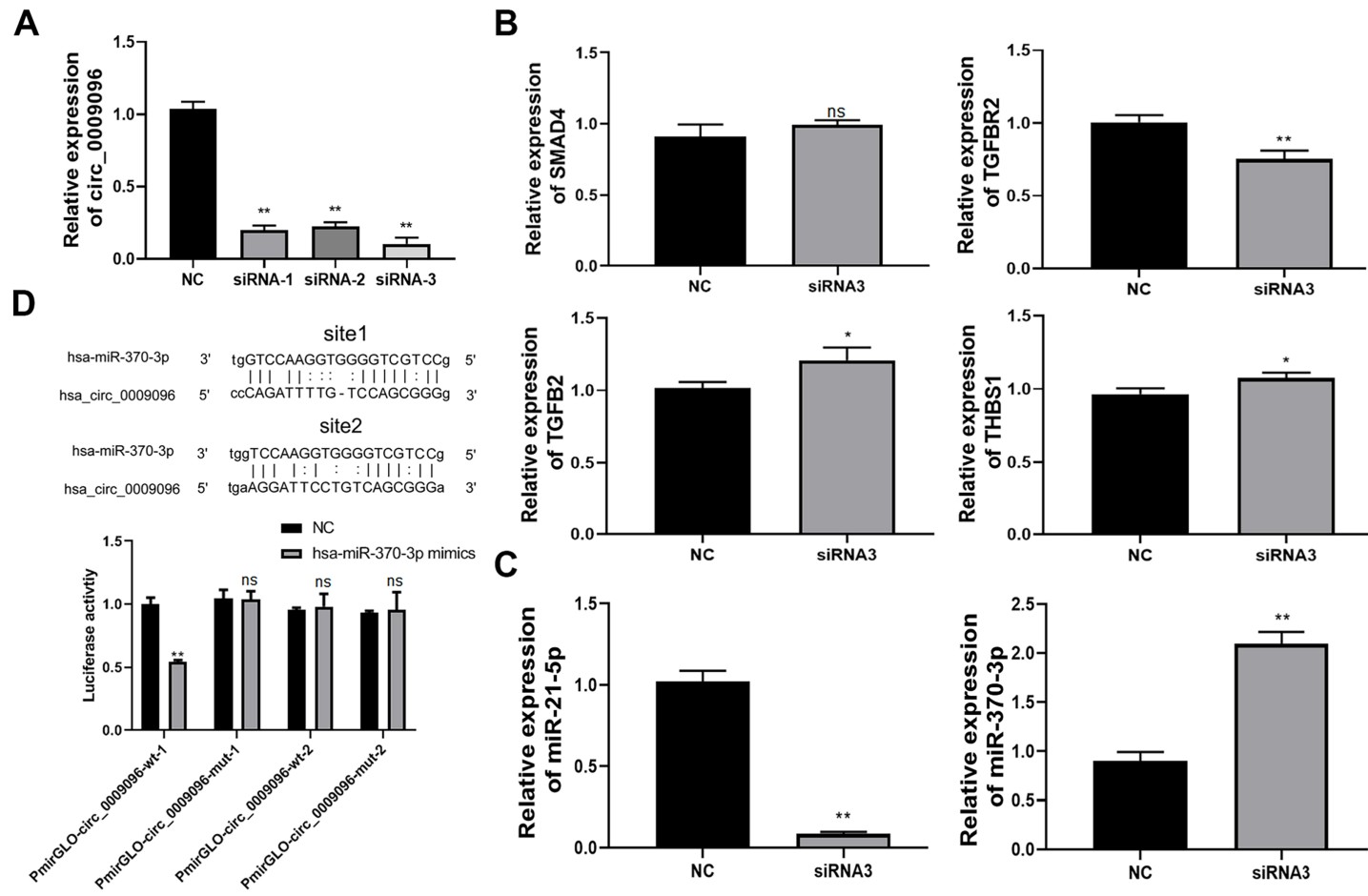

**Figure 3 Hsa_circ_0009096 sponges miR-370-3p to modulate TGFBR2 expression in LX-2 cells.** (A) LX-2 cells were transfected with hsa_circ_0009096 siRNAs for 24 h and then RT-qPCR was used to assess the interference effect of siRNAs ($n$ = 3 biological replicates per group). (B) Relative mRNA expression of potential target genes of hsa_circ_0009096 ( SMAD4, TGFB2, TGFBR2, and THBS1 ) in LX-2 cells after transfection with hsa_circ_ 0009096 siRNA3 for 24 h were detected *via* RT-qPCR ($n$ = 3 biological replicates per group). (C) The expression of miR-21-5p and miR-370-3p in LX-2 cells transfected with hsa_circ_ 0009096 siRNA3 for 24 h were analyzed by the RT-qPCR ($n$ = 3 biological replicates per group). (D) The interaction between hsa_circ_0009096 and miR-370-3p in 293T cells was confirmed by the dual luciferase reporter assay ($n$ = 3 biological replicates per group). NC: negative control; TGFBR2: TGF beta receptor 2; SMAD4: Mothers Against Decepentaplegic Homolog 4; TGFB2: Transforming Growth Factor Beta 2; THBS1: Thrombospondin 1; NS, not significant ($p$ > 0.05), *$p$ < 0.05, **$p$ < 0.01.

in the LX-2 cells after TGF-β1 treatment (Fig. 4F). Using ICC, we confirmed that the a-SMA and COL1A1 protein levels in the LX-2 cells increased significantly after TGF-β1 treatment, which were then significantly mitigated by hsa_circ_0009096 siRNA (Fig. 4G). Consistent with the RT-qPCR results, the WB results showed that interference with hsa_circ_0009096 reversed the TGF-β1-induced upregulation of COL1A1, a-SMA, and TGFBR2 protein expression (Fig. 4H). These results revealed that hsa_circ_0009096 could promote TGF-β1-induced HSC proliferation and liver fibrosis by inhibiting miR-370-3p expression and upregulating *TGFBR2* expression.

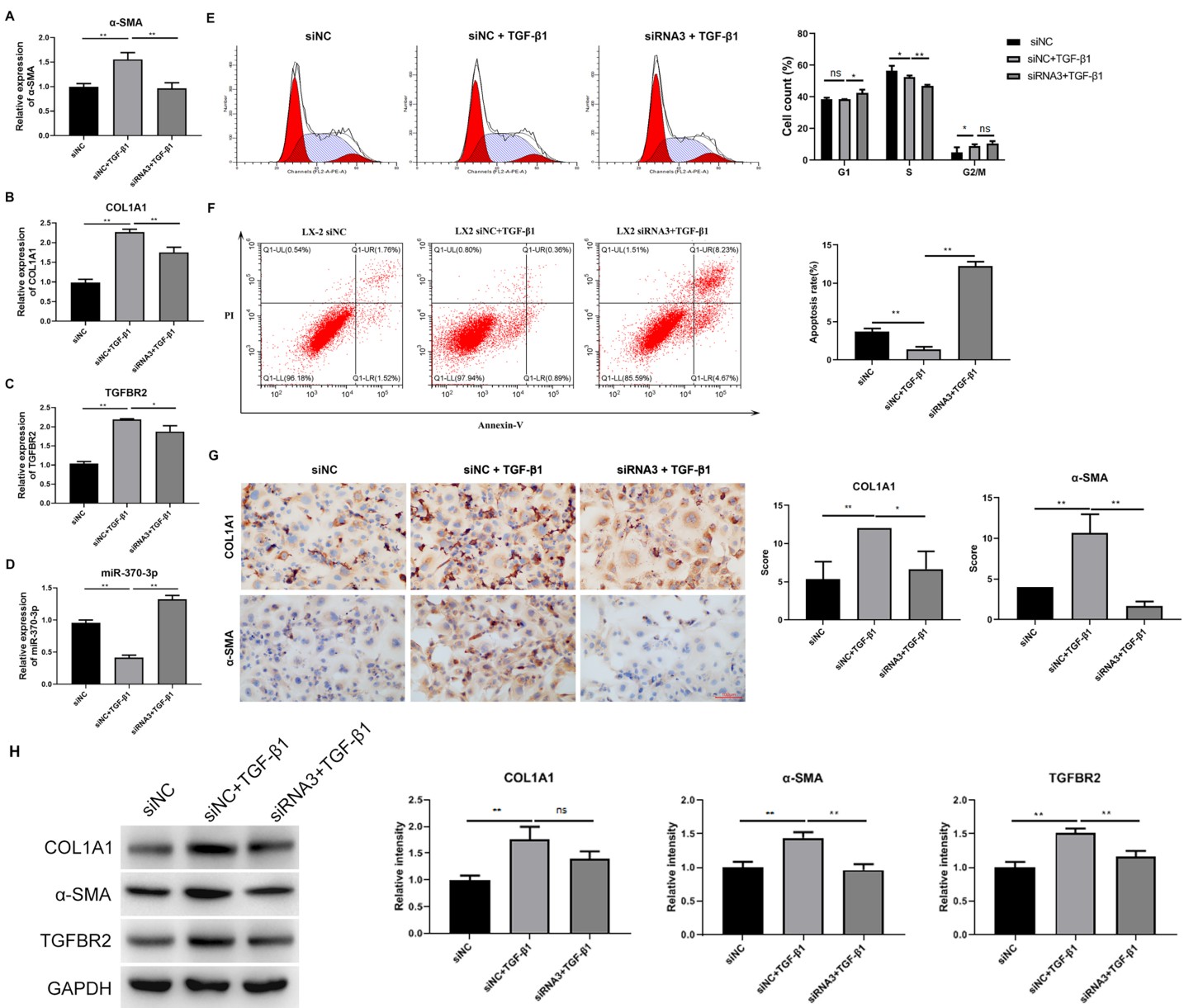

**Figure 4 Hsa_circ_0009096 silencing suppressed TGF-β1-induced HSC proliferation and fibrosis.** LX-2 cells were transfected with siNC or hsa_circ_0009096 siRNA3 for 24 h, followed by treatment without or with 2 ng/mL rhTGF-β1 for 24 h. (A–D) The miRNA/mRNA expression of α-SMA, COL1A1, TGFBR2, and miR-370-3p were detected by RT-qPCR ($n$ = 3 biological replicates per group). (E–F) The cell cycle progression and the apoptosis of LX-2 cells were assessed by flow cytometry ($n$ = 3 biological replicates per group). (G) Alterations of α-SMA and COL1A1 protein abundances in LX-2 cells were evaluated by Immunocytochemistry ($n$ = 3 biological replicates per group). (H) The protein expression of TGFBR2, COL1A1, and α-SMA was examined by western blotting ($n$ = 3 biological replicates per group). NC: negative control; TGFBR2: TGF beta receptor 2; COL1A1: collagen 1A1; α-SMA: alpha-smooth muscle actin; NS, not significant ($p > 0.05$), *$p < 0.05$, **$p < 0.01$.

### MiR-370-3p mediated hsa_circ_0009096-regulated proliferation and fibrosis of TGF-β1-treated HSCs

To study the potential mediating effects of miR-370-3p in HSC proliferation and fibrosis regulated by hsa_circ_0009096, we suppressed miR-370-3p expression in the LX-2 cells

transfected with hsa_circ_0009096 siRNA by treatment with miR-370-3p inhibitors. The miR-370-3p inhibitor significantly inhibited the hsa_circ_0009096 siRNA-induced elevation of the miR-370-3p expression in the LX-2 cells (Fig. 5A). In contrast, hsa_circ_0009096 siRNA repressed the expression of TGFBR2, a-SMA, and COL1A1 in the LX-2 cells, which were all significantly recovered by miR-370-3p inhibitors (Figs. 5B–5D). Furthermore, the LX-2 cell cycle arrest at the G1 stage induced by hsa_circ_0009096 siRNA transfection was significantly alleviated by miR-370-3p inhibitor treatment (Fig. 5E). Similarly, the increase in apoptotic LX-2 cells caused by hsa_circ_0009096 siRNA transfection was also remarkably mitigated by treatment with miR-370-3p inhibitors (Fig. 5F). Finally, our ICC assay showed that the miR-370-3p inhibitors significantly alleviated the hsa_circ_0009096 siRNA-induced downregulation of the a-SMA and COL1A1 protein levels in the LX-2 cells (Fig. 5G). Together, these assays proved that the regulation of HSC proliferation and fibrosis by hsa_circ_0009096 was mediated by its binding with miR-370-3p to modulate *TGFBR2* expression.

## DISCUSSION

BA pathogenesis is mediated by the abnormal activation and transformation of HSCs, which commonly leads to HSC proliferation, ECM deposition, and liver fibrosis (*Xiao et al., 2014*; *Shen et al., 2019*; *Vij & Rela, 2020*). Currently, the mechanisms driving HSC activation and hepatic fibrosis underlying BA development remain poorly elucidated.

In the past decades, circular RNAs have been established as potent drivers of various human disorders (*Akhter, 2018*; *Beltrán-García et al., 2020*), but little is known about their implication in BA progression. In this study, we established a cellular model by treating LX-2 cells with TGF-β1 to explore the cellular functions and molecular mechanisms of hsa_circ_0009096 in regulating the activation of HSCs and hepatic fibrosis associated with BA development. We discovered that hsa_circ_0009096, a transcription product of the *UTRN* gene, is located at chr6: 144772505–144780490, and exhibited significant expressional elevation in TGF-β1-treated HSCs, consistent with our previous analysis in clinic samples from patients with BA (*Zhang et al., 2023*) and the same cell model (*Liu et al., 2023*). More importantly, we found that hsa_circ_0009096 silencing effectively repressed HSC cycle progression and promoted their apoptosis under TGF-β1 treatment, accompanied by lowered expression of the ECM biomarkers a-SMA and COL1A1. These results provide convincing data suggesting hsa_circ_0009096 as one new regulator of HSC activation and liver fibrosis during BA development.

As mentioned above, the widespread biological functions of circRNAs have been mainly attributed to one molecular mechanism, namely, their interaction with microRNAs as decoys to modulate target gene expression (*Seimiya et al., 2020*; *Zhou et al., 2020*). In our previous study, we found that hsa_circ_0009096 could act as a ceRNA molecule adsorbing multiple miRNAs targeting to regulate the expression of multiple target genes in the TGFβ signaling pathway using bioinformatics prediction (*Zhang et al., 2023*). MiR-370-3p is one of them. In this study, we validated the association of hsa_circ_0009096 with miR-370-3p in the 293T cells using the dual luciferase reporter method. Notably, miR-370-3p expression was significantly upregulated in the LX-2 cells after transfection with

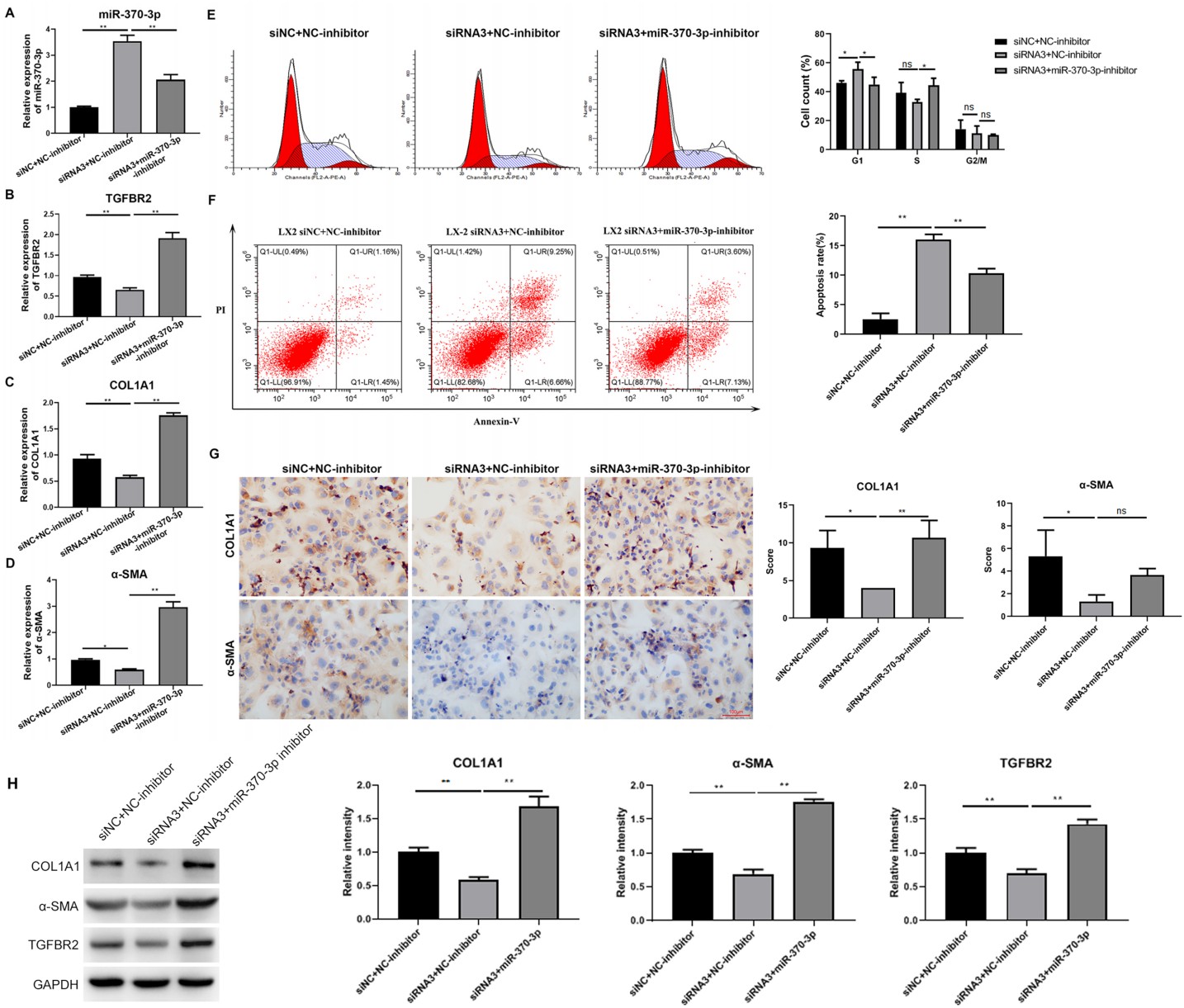

**Figure 5 MiR-370-3p mediated hsa_circ_0009096-regulated proliferation and fibrosis of TGF-β1-treated HSCs.** LX-2 cells were transfected with siNC+NC-inhibitor, hsa_circ_0009096 siRNA3+NC-inhibitor, and hsa_circ_0009096 siRNA3+miR-370-3p-inhibitor, respectively, for 24 h, followed by treatment with 2 ng/mL rhTGF-β1 for 24 h. (A–D) The miRNA/mRNA expression of miR-370-3p, TGFBR2, COL1A1, and a-SMA were detected by RT-qPCR ($n = 3$ biological replicates per group). (E–F) The cell cycle progression and the apoptosis of LX-2 cells were assessed by flow cytometry ($n = 3$ biological replicates per group). (G) Alterations of a-SMA and COL1A1 protein abundances in LX-2 cells were evaluated by Immunocytochemistry ($n = 3$ biological replicates per group). (H) The protein expression of TGFBR2, COL1A1, and a-SMA was examined by western blotting ($n = 3$ biological replicates per group). NC: negative control; TGFBR2: TGF beta receptor 2; COL1A1: collagen 1A1; α-SMA: alpha-smooth muscle actin; NS, not significant ($p > 0.05$), *$p < 0.05$, **$p < 0.01$.

hsa_circ_0009096 siRNA. This could be because circRNA acts as an endogenous miRNA sponge that adsorbs miRNAs and inhibits miRNA transcription, processing, or degradation, rather than inhibiting miRNA activity (*Singh, Sinha & Panda, 2023*). Importantly, we persuasively verified the roles of miR-370-3p in mediating

hsa_circ_0009096-regulated LX-2 cell proliferation and fibrosis by transfecting miR-370-3p inhibitors into the LX-2 cells with silenced hsa_circ_0009096 expression. This evidence has characterized miR-370-3p as a critical microRNA target of hsa_circ_0009096 in HSC activation during BA development.

TGF-β-induced signaling events also play essential roles in hepatic fibrosis. For instance, a recent study showed that TGF-β could promote liver fibrosis by directly activating Janus kinase 1 (JAK1) and signal transducer and activator of transcription 3 (STAT3), with the assistance of the SMAD signaling pathways (*Tang et al., 2017*). In the TGF-β signaling cascade, the TGFBR2 is an essential transmembrane serine/threonine kinase and serves as a critical component of the TGF-beta receptor, which could be targeted by multiple microRNAs under different contexts according to recent investigations (*Fu et al., 2020*; *Zhang et al., 2020*). More importantly, a recent study in cashmere goats found that miR-370-3p could directly target and repress the expression of *TGFBR2* and Fibroblast Growth Factor Receptor 2 (*FGFR2*) to modulate epithelial cell and fibroblast proliferation and hair follicle morphogenesis in cashmere goats (*Hai et al., 2021*). In our study, *TGFBR2* was also predicted *via* bioinformatics as a target gene downstream of miR-370-3p, which may be involved in HSC activation and BA development. To validate this, we observed a significantly decreased *TGFBR2* expression in the LX-2 cells that were transfected with hsa_circ_0009096 siRNA, as well as in the cellular hepatic fibrosis model induced by TGF-β1 treatment. Additionally, the decreased *TGFBR2* expression in the LX-2 cells transfected with hsa_circ_0009096 siRNA was greatly recovered by miR-370-3p inhibitors. These results suggested that the modulation of HSC proliferation and hepatic fibrosis by the hsa_circ_0009096/miR-370-3p axis may be mediated by targeting *TGFBR2* expression.

## CONCLUSIONS

In summary, we demonstrated in this study, using a cellular model based on the LX-2 cells, that the hsa_circ_0009096, which is highly expressed in hepatic tissues of patients with BA, could fundamentally promote HSC proliferation, activation, and hepatic fibrosis, through the sponging of miR-370-3p to modulate *TGFBR2* expression and the TGF-β1 signaling pathway. These investigations provided new insights into the molecular mechanisms underlying BA pathogenesis, which could be applied as a basis for developing diagnostic and therapeutic methods for the clinical management of BA. However, our study was limited by the use of a single cell line, and the specific regulatory mechanism by which hsa_circ_0009096 inhibits miR-370-3p expression was not demonstrated. We recognize these limitations and will work to improve upon them.

### Funding

The study was supported by the Guangdong High-level Hospital Construction Fund, the Sanming Project of Medicine in Shenzhen (grant number SZSM201812055), the Pediatric

nutrition support team develops scientific research projects (grant number xm-2019-000-0114-03) and the Research topic on academic and postgraduate education in China (grant number 2020MSA126). The funders had no role in study design, data collection and analysis, decision to publish, or preparation of the manuscript.

## Grant Disclosures

The following grant information was disclosed by the authors:
Guangdong High-level Hospital Construction Fund.
Sanming Project of Medicine in Shenzhen: SZSM201812055.
Pediatric nutrition support team develops scientific research projects: xm-2019-000-0114-03.
Research topic on academic and postgraduate education: 2020MSA126.

## Competing Interests

The authors declare that they have no competing interests.

## Author Contributions

- Zhouguang Wu conceived and designed the experiments, performed the experiments, prepared figures and/or tables, authored or reviewed drafts of the article, and approved the final draft.
- Bin Wang conceived and designed the experiments, authored or reviewed drafts of the article, and approved the final draft.
- Siqi Chen performed the experiments, prepared figures and/or tables, authored or reviewed drafts of the article, and approved the final draft.
- Taoyan Zuo performed the experiments, prepared figures and/or tables, and approved the final draft.
- Wenjie Zhang performed the experiments, prepared figures and/or tables, and approved the final draft.
- Zhen Cheng analyzed the data, prepared figures and/or tables, and approved the final draft.
- Jingru Fu analyzed the data, prepared figures and/or tables, and approved the final draft.
- Jiafeng Gong analyzed the data, prepared figures and/or tables, and approved the final draft.

## Data Availability

The raw measurements are available in the Supplemental Files.

## Supplemental Information

Supplemental information for this article can be found online at http://dx.doi.org/10.7717/peerj.17356#supplemental-information.

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
