# Peer review of "Hsa_circ_0009096/miR-370-3p modulates hepatic stellate cell proliferation and fibrosis during biliary atresia pathogenesis"

_PeerJ, doi:10.7717/peerj.17356_

## Round 0.1 · original submission · Major Revisions

The manuscript has received feedback from three reviewers.

Reviewer 1 focuses on concerns related to the experimental design, suggesting improvements in the relevance of the in vitro model, data presentation, proof of circular RNA existence, and completeness of the mechanistic study.

Reviewer 2 highlights language issues, concerns about PCR primer specificity, and the potential impact on the validity of the findings.

Reviewer 3 raises concerns about coherency, rationale, the explanation of sponging activity, and the need for validation at translational levels.

The common themes across the reviews include the need for improved clarity, justification, and validation, as well as addressing language-related issues. The editor's report emphasizes the importance of comprehensive revisions to address these concerns and enhance the overall quality of the manuscript.

**Language Note:** The review process has identified that the English language must be improved. PeerJ can provide language editing services - please contact us at copyediting@peerj.com for pricing (be sure to provide your manuscript number and title). Alternatively, you should make your own arrangements to improve the language quality and provide details in your response letter. – PeerJ Staff

Reviewer 1 ·

Basic reporting

In this study, Wu et al. aim to establish the role of circular RNA 0009096 (Hsa_circ_0009096) in hepatic stellate cell (HSC) activation in biliary atresia (BA) pathogenesis. Although the function of this circular RNA is relatively unknown and hence presents a novel aspect of the study, the experimental design and rationale of the study are poorly executed. I have drafted comments for authors in the next section to consider for improvement.

Experimental design

1. How is the in vitro model of HSC specifically relevant to BA pathogenesis? The activation of HSCs relies on TGF-beta, which is a common trigger for fibrosis found in all chronic liver diseases.
2. The in vitro study that employs a single cell line may have concerns for reproducibility.
3. The authors should present the data on Hsa_circ_0009096 expression levels in normal and BA liver samples. In addition, how do the authors assume that the altered expression of Hsa_circ_0009096 arises from HSCs? Why do the authors specifically investigate HSCs for in vitro experiments?
4. The authors do not prove the existence of Hsa_circ_0009096 in HSCs via RNaseR and Actinomycin D treatments.
5. A circular RNA can sponge multiple microRNAs (miRNAs). It is unclear why the authors specifically chose to focus on two miRNAs. In addition, how many binding sites of each miRNA appear within Hsa_circ_0009096? Is it sufficient for the sponging activity?
6. What are the direct target(s) of miR-21-5p and miR-370-3p in the TGF-beta pathway? Do any of them directly regulate TGF-beta receptor 2? The authors did not show these data to complete the mechanistic study.
7. Validation of gene expression at the protein level is required for all experiments. Quantitative analysis should be included.
8. Quantification of the cell cycle analysis is required for the study.

Validity of the findings

See above.

Reviewer 2 ·

Basic reporting

The usage of technical terminology is inconsistent, and there are noticeable grammatical and spelling errors. It requires revision by a fluent English speaker to reach a professional standard.
The manuscript provides a reasonable background and context in its field, with adequate literature references. However, the depth and breadth of the discussion could be enhanced to strengthen the manuscript's contribution to the field.
The choice of PCR primers in the study raises significant concerns. A BLAST analysis reveals that the primers used are not unique and can match multiple circRNAs, which questions the specificity and reliability of the results presented.
Given the issues with the PCR primers, there is a justified skepticism towards the study's conclusions. The potential lack of primer specificity undermines the validity of the results and their relevance to the hypotheses, casting doubt on the overall findings of the study.

Experimental design

The study presents original research that is within the journal's scope. However, it could further emphasize how it advances the field to fully meet the journal's aims.
The research question is clearly defined and relevant. The study explains how it fills a knowledge gap, but the significance of this contribution could be more impactful.
The investigation is rigorous, demonstrating high technical and ethical standards. However, there might be room for further refinement to reach exceptional standards.
The methods are detailed and provide enough information for replication. Still, additional clarity or elaboration in certain areas could enhance the methodological robustness.

Validity of the findings

Due to issues with the PCR primers' specificity, the conclusions drawn are significantly undermined. Although they are well-stated and linked to the research question, the validity of these conclusions is questionable, as they are based on potentially unreliable results. This impacts the overall credibility of the study's findings.

Additional comments

1, The introduction section of the manuscript should focus more on providing a detailed background of Hsa_circ_0009096 specifically, rather than listing various other circRNAs. This would give readers a better understanding of the specific circRNA under study and its relevance to the research being presented.
2, Verifying the effectiveness and uniqueness of the Hsa_circ_0009096 primers through multiple methods is crucial for the integrity of the paper's conclusions. This verification process is vital to ensure the accuracy and reliability of the results related to Hsa_circ_0009096, which in turn, has a decisive impact on the overall validity and credibility of the study's findings.
3, The title at line 178 of the manuscript, "Cell apoptosis," may not be the most suitable or descriptive for the content it covers. It might benefit from a more specific or detailed title that accurately reflects the scope and focus of the section it introduces.
4, Bioinformatics analyses predict that Hsa_circ_0009096 can potentially bind with multiple miRNAs. What led to the specific focus on miR-370-3p in your study?
5, In your paper, you mention that Hsa_circ_0009096 acts as a sponge to adsorb miR-370-3p. Typically, the sponging action does not alter the expression of miRNA. However, in your study, there's a reduction in miR-370-3p expression. In your discussion, it would be important to address the potential mechanisms that might explain this observation.
6, I've reviewed the manuscript and can confirm that Figure 2B does not have P-values marked. It's important for statistical significance to be clearly indicated in such figures to enable readers to understand the strength and reliability of the findings.
7, The statistical analysis for Figure 3G, FIG 4 G in the manuscript has not been performed or is not evident in the provided document. For a comprehensive and rigorous scientific study, it's crucial to conduct appropriate statistical analysis on experimental data. This ensures that the results are statistically significant and reliable, thereby supporting the study's conclusions.

Reviewer 3 ·

Basic reporting

The coherency is not maintained in the “introduction and discussion parts” of the manuscript. Sudden deviations are frequent and so presentation needs to be streamlined and crisp

Rationale of the study and overall experimental procedures seems not justifiable especially in context of primers and circRNAs.

Experimental design

The potential molecular mechanism pertaining to Hsa_circ_0009096 and Hepatic stellate cells needs to be explained.

Explanation for sponging activity needs to be addressed

For authentication, validation of transcriptional results at translational levels should also be provided for all experiments

Validity of the findings

statistical analysis needs revision.

---

## Round 0.2 · Minor Revisions

Overall, the findings have been strengthened by new results, as noted by Reviewers. The manuscript appears to be in good shape. for further consideration pending final editorial review.

However, I would request the authors to add details of technical and biological replicate numbers in all figures and sub-figures, and show all blots for all replicate experiments, not just the ones presented, before the next steps.

Reviewer 1 ·

Basic reporting

I thank the authors for addressing my concerns. I understand that some experiments cannot be achieved in this study. I hope the authors will pursue this in their future works.

Experimental design

The revised version is fine.

Validity of the findings

The findings are strengthened by some new results.

Reviewer 2 ·

Basic reporting

No comments

Experimental design

No comments

Validity of the findings

No comments

Additional comments

The authors addressed all my concerns, the paper is appropriate to be published.

---

## Round 0.3 · accepted · Accept

The authors have addressed the concerns raised by both reviewers, with Reviewer 1 appreciating the authors' efforts in clarifying and understanding the limitations of their study. Reviewer 1 also expressed hope that the authors would pursue the unattainable experiments in future works. Reviewer 2 had no comments on the basic reporting, experimental design, or validity of the findings, indicating satisfaction with the revisions made by the authors. Both reviewers deemed the paper appropriate for publication. Based on the reviewers' comments and the authors' responses, it is recommended that the paper be accepted for publication.